# Use of Information and Communication Technologies among Adults in Weight Control: Systematic Review and Meta-Analysis

**DOI:** 10.3390/nu14224809

**Published:** 2022-11-14

**Authors:** Thatiana Wanessa Oliveira, Priscilla Perez da Silva Pereira, Leonice Antunes Fonseca, Luna Mares Lopes de Oliveira, Dauster Souza Pereira, Carla Paola Domingues Neira, Ana Claudia Morais Godoy Figueiredo

**Affiliations:** 1Health Care Research Laboratory, Federal University of Rondônia, Br 364, Porto Velho 76801-059, Brazil; 2Centro Interdisciplinar de Novas Tecnologias na Educação (CINTED), Federal Institute of Rondônia, Avenida Lauro Sodré, 6500, Porto Velho 76804-124, Brazil; 3Federal District Health Department, SRTVN, 701 Norte, Brasília 70390-125, Brazil

**Keywords:** Body Mass Index, information technology, meta-analysis

## Abstract

Information and communication technologies are part of our day-to-day life in the execution of all activities, including health care. However, it is not known how much the use of technologies can contribute to the adoption of healthy lifestyle habits. Thus, the objective of this study was to analyze whether the use of information and communication technologies contributes to weight control among adults when compared to the traditional approach method. The search was performed in November 2021 in eight electronic databases in addition to gray literature bases. The quality of the studies was assessed using the Cochrane risk of bias tool. The standardized mean difference was used for the meta-analytic measurement using the random effects model using the Dersimonian–Laid method in the Stata statistical package version 17. The body mass index of the intervention group decreased by an average of 0.56 (95% CI: −0.83; −0.30) when compared to the control group. When comparing the before and after groups, the intervention group also had a greater reduction in BMI (summarized mean: −0.83; 95% CI: −1.40; −0.26). Information and communication technologies contribute to the reduction of the body mass index in the adult population when compared to the traditional model of monitoring. Prospero registration: number 42020186340.

## 1. Introduction

Information and communication technologies have provided new lifestyles, consumption, teaching, learning, relationships with professionals and health services [1,2]. Interventions to solve health problems, especially those caused by inadequate habits such as hygiene, food and physical activity, are possible themes as the focus of promotion actions based on interventions mediated by technologies [3,4].

A relevant subject to be discussed and worked on nowadays is food education. The constant increase in the consumption of ultra-processed foods with little nutritional value and to which people have easy access, is a cause for concern; the increase in the consumption of ultra-processed foods by the population can cause future health problems due to the low nutritional value of these foods [5,6].

A healthy diet among adults helps to protect against malnutrition, favors food security, and acts against chronic non-communicable diseases, including diabetes, cardiovascular diseases, stroke, and cancer [7,8].

An integrative literature review, conducted in 2019, included eight studies on the use of digital technologies to promote healthy eating habits among adolescents. The technologies studied were games, websites and programs, and among the changes found with their use, there was a greater consumption of fruits and vegetables and greater practice of physical activity; the study concluded that the use of technologies promoted changes in the lifestyle and healthy eating habits, and that the technology must adapt to social and economic factors in its implementation and use [9].

A systematic review of 88 studies verified the effectiveness of interactive social media interventions among adults and the changes in health behaviors. For changes in body weight, four studies were included that found a decrease in body mass index [10]. However, in these studies, the use of information and communication technologies were not evaluated as part of a weight loss treatment.

Considering that information and communication technologies have been increasingly used in educational processes, including health areas, it is important to investigate the capacity of these resources in generating lifestyle changes. No systematic review was found that evaluated the impact of information and communication technologies on changing eating habits, with the final outcome being changes in body weight among adults. Thus, the objective of this study was to analyze whether the use of information and communication technologies contributes to weight control among adults when compared to the traditional approach method—face-to-face (presential) and without the use of information and communication technologies.

## 2. Method

The systematic review study was registered in the Prospective International Registry of Systematic Reviews (PROSPERO) under the Center for Reviews and Dissemination (CRD) number 42020186340. The protocol design was performed according to the statement of Preferred Reporting Items for Systematic Reviews and Meta-Analyses [11] (Appendix A).

### 2.1. Eligibility Criteria

Studies of randomized clinical trials that compared the use of information and communication technologies in the development of healthy eating habits and conventional treatment in weight control among adults from 18 to 65 years were considered eligible. Any type of technology that used cell phones, smartphones, computers and software was considered as information and communication technologies. As a traditional approach, the service was exclusively face-to-face and without the use of information and communication technologies.

The BMI, also called the Quételet index, was used as a measure for assessing weight control. This measure is calculated by dividing body mass in kilograms by the square of height in meters, it is an indicator of the nutritional status of adults. BMI is classified as: (a) less than 18.5: underweight; (b) 18.5 and 24.9: normal weight; (c) 25 and 29.9: overweight; (d) equal to or above 30: obesity [12].

Body weight was chosen as the outcome because the assessment of the population’s body composition is the most-used method in the population’s nutritional diagnosis, as it is cheap, non-invasive, universally applicable and well accepted by the population [13,14].

Studies with pregnant or postpartum women, participants with terminal illnesses such as cancer, pilot studies, and studies in which technological resources were used in both groups were excluded.

### 2.2. Information Search, Query Strategy and Selected Studies

The search was performed in November 2021 in the following databases: Medline, Embase, Central, LILACS, SciELO, Web of Science, Scopus and PsychInfo using the search strategy considering the controlled vocabularies Medical Subject Headings (MeSH) and Emtree, free terms related to the topic, as well as their synonyms, keywords and using Boolean operators to combine the descriptors (Appendix A). The Peer Review of Electronic Search Strategies (PRESS) tool was used to qualify the search strategy by an independent reviewer. There was no restriction for language or publication date.

A search for references in the selected articles was also carried out, in the gray literature in the bases, including: OpenGrey, ProQuest, National Technical Information Service (NTIS), MedRxiv, BiorXiv, and WHO Library Database. The following clinical trial registry databases were consulted: clinicaltrials.gov, International Clinical Trials Registry Platform, The European Union Clinical Trials Register, Brazilian Registry of Clinical Trials (ReBEC).

For the study selection process, the Rayyan platform was used. Initially, a calibration of the selection was carried out with the reviewers to confirm their understanding of the eligibility criteria. After excluding duplicate studies, two reviewers assessed the title and abstract for full reading—Oliveira, T.W.; Oliveira, L.M.L.; Andrade, L.A. Disagreements were resolved by consensus or by consulting another reviewer—Pereira, P.S.P.

### 2.3. Data Extraction

Data was extracted by Oliveira, T.W.; Pereira, P.P.S.; Pereira, D.S. in a standardized spreadsheet, including the following data: title, first author and year, parents, sample size, mean age, sex, marital status, education, race, focus of intervention, intervention time, BMI in the control group and in the intervention group before and after the actions. For studies with information with standard deviation values, the authors were contacted requesting this information.

### 2.4. Methodological Quality

The quality of the studies was assessed using the Cochrane risk of bias tool. This tool is composed of two parts, which contain seven domains, namely: random sequence generation, allocation concealment, blinding of participants and professionals, blinding of outcome assessors, incomplete outcomes, selective outcome reporting and other sources of bias. For each of these domains, the risk of bias is assessed, being classified as high, uncertain (nuclear) or low risk of bias [15].

### 2.5. Data Analysis

The data analysis process was performed using the program STATA^®^ version 17 (StataCorp LLC, College Station, TX, USA), serial number: 301706385466. The degree of statistical heterogeneity of the studies was evaluated using the I-Square (I2), applying the following cut-off points: 0 to 40% not very important; 30 to 60% moderate; 50 to 90% substantial; and 75 to 100% considerable [15]. The standardized mean difference was used for the meta-analytic measurement of the evaluated results. To estimate statistical significance, 95% confidence intervals were calculated. The random effects model was used through the Dersimonian–Laid method. The random effects model was chosen to adopt a more conservative approach, since statistical heterogeneity was considered high for one of the meta-analyses. Furthermore, high methodological heterogeneity was perceived between the studies. It was not possible to measure publication bias as fewer than 10 studies were included in the meta-analyses [15].

### 2.6. Results

Nine-hundred and seventy-nine studies were identified from the databases. Of these, 124 duplicates were excluded, resulting in 16 studies included in the present study, with a sample of 6907 adults (Figure 1). The studies were carried out in the following continents: Asia [16,17], Americas [18,19,20,21,22,23], Europe [17,24,25,26,27,28,29,30] (Table 1). Only one study was published in the 2000s [29]. Nine studies were published from 2012 to 2015 [17,19,20,21,22,26,27,30,31] and six studies were published from 2016 to 2020 [16,18,23,24,25,28]. The sample size of the included studies ranged from 32 to 1790, and some had samples smaller than 100 participants [17,24,27,29,31].

Regarding gender, only one study had a sample composed of a majority of men [22], and one study had only women [31] and another study had only men [29]. The mean age of the study ranged between 24 and 58 years. The intervention time had a minimum of three months [17,20,24,25,27,28,30] and a maximum of 21 months [18]. Of the 16 studies included, only five provided information on schooling, one mentioned the highest level of education being high school [26], in another study the intervention group was mostly at the levels of high school and higher education in the control group [21,26], in the other studies the most frequent level was higher education [16,22,24]. One of the studies was carried out among university students [20].

The focus of information and communication technology interventions were: weight loss, diet, eating behavior change, physical activity, recipes, personalized dietary advice, goals, personalized feedback, eating disorders, body satisfaction, healthy lifestyle, diabetes prevention and its implications, reducing blood glucose, understanding which foods are rich in fat, sugar and salt, reducing fat intake, and increasing consumption of fruits and vegetables.

Among the technologies that were used in the intervention, they are divided into: weekly sending of texts, images and video by Short Message Service (SMS), e-mail or social network applications (for example, WhatsApp); games; websites with educational programs; web conferences; or virtual reality. In the control group, the strategies used were: standard care, performed in an office or hospital, by doctors and other professionals. Only one study indicated that follow-up was carried out specifically by a nutritionist [18]; another study mentions that it provided orientation based on the guidelines and that patients received attention in primary care. One study mentions that medical care was provided with nutritional guidelines [27]; another study [16] mentions that informative pamphlets were used in the guidelines.

Seven studies presented information that made it possible to carry out the meta-analysis. It was possible to verify that the BMI of the group that received the interventions decreased by an average of 0.56 (95% CI: −0.83; −0.30) and presented low heterogeneity between studies (Figure 2). When comparing the groups before and after the interventions, the group that participated in the actions through information and communication technologies also had a greater reduction in BMI (mean: −0.83; 95% CI: −1.40; −0.26) with moderate heterogeneity (I:64.95%) (Figure 3).

Regarding the evaluation of the methodological quality of the studies, the risk of blinding bias of the participants and evaluators was classified as low, as it was considered that in no intervention it was possible to have blinding (Figure 4). Biases introduced due to incomplete outcome and selective reporting were assessed as high in 20% of studies. In general, the studies were evaluated as having good methodological quality (most items had a higher frequency for low and uncertain risk of bias).

## 3. Discussion

Information and communication technologies contributed to greater weight loss when compared to conventional interventions—face-to-face only. The most performed actions were courses divided into online modules, sending SMS, and food control through a calorie-counting application. The focus of interventions was weight loss, increased healthy eating, individualized dietary advice, prevention of eating disorders, improved body image, diabetes and its implications, lowering blood glucose, and participation in physical activity.

Most studies were conducted in groups with an average age of over 40 years, which shows that the use of information and communication technologies can be a good option for health education, health self-management and monitoring by health professionals in older adults. old. Only five studies presented the level of education and most had high school or higher education. A higher level of education was related to a higher level of food quality. Low schooling was related to worse quality of life, higher prevalence of non-communicable chronic diseases, mental disorders and depression. A study conducted in one in Brazil, in 2018, found that people with twelve or more years of schooling had a prevalence of fruit and vegetable consumption of 38% against 23% of individuals with up to eight years of schooling [32].

Weight losses or dietary changes are always greater after the intervention and tend to decrease after that period. In a meta-analysis of four studies with behavioral interventions, including with 301 participants who followed the interventions for six months, showed a favorable effect on lowering BMI in children younger than 12 years of age who received parent-focused interventions compared to usual practices. However, at 12 months of follow-up, the meta-analysis of three studies, including 264 participants, did not show a significant reduction in BMI, but there were significant changes in the dietary pattern [33].

In our meta-analysis, we found some studies indicating that there was weight loss in the first six months and after that period, this weight remained stable [23]. Two other studies reported that there was weight loss in the first six months and it was not maintained at 12 months, but there were significant improvements in food quality [26,27].

Among the studies that reported that they had the help of feedback, either by SMS or calls, six studies in this review reported that they used these tools to increase adherence to the technology used [16,17,21,22,23,28]. The experience of constantly receiving feedback reinforces that adherence to the use of technology increases self-assessment and monitoring of diet and physical activity [17].

In this review, six studies were found that claimed to use a psychological approach in the intervention [19,20,23,24,26,31]. It is known that eating behavior is related to the psychological aspects of food intake and how the behavior impacts on consumption, so it is preferable that the approach goes beyond nutritional behavior [34].

Virtual environments help in the development and training of technological skills, bringing many benefits to those who use them, but in addition to autonomy, it also increases the responsibility for their own performance [35]. In this review, a study developed a web-based application that aimed to monitor diet and physical activity while instructing and encouraging healthy diet and physical activity [17].

An effective intervention requires greater efficiency of mobile technology, social support and human interaction, in addition to an adaptive approach, that is, according to the profile of the user [36]. The studies presented show the importance of combining the use of technologies for a more efficient and effective health care.

One of the limitations identified was related to the lack of detail on the type of care that the control group received in some of the studies. Importantly, the results of interventions to reduce BMI may vary with age, due to differences in metabolism and nutritional needs. However, in general, it was found through meta-analysis that information and communication technologies can be a good tool in self-management of weight control through health education and used as support for face-to-face care for health professionals. Another limitation of this study is that among the studies included in the meta-analysis, it was not possible to measure publication bias, as fewer than 10 studies were included.

As a strength of this study, the rigor in its conduct can be cited: search of sensitive literature, no publication language or date restrictions, inclusion of a search of gray literature and selection of studies, data extraction and methodological evaluation carried out independently by at least two authors. The present systematic review followed PRISMA.

## 4. Conclusions

This study found that information and communication technologies can contribute to the reduction of BMI in the adult population when compared to the traditional follow-up model. The use of technologies has been the agenda of health policies, with the aim of reducing the rates of diseases that depend on eating behaviors and that can be modified throughout life, and that in addition to weight loss, there is a change in eating behavior.

Therefore, the results of this review provide answers to the questions of this research and seek to promote the development of research in the area of public health with the production of educational technologies and evaluation of their impact with the adult public, contributing to policies of nutrition and adequate and healthy food.

The use of technologies as an adjunct to conventional care should still be investigated. Given the current moment that the world is going through, the pandemic caused by the new coronavirus, it is important to replicate this research after this period to verify the impact of information and communication technologies on changing lifestyle habits.

## Figures and Tables

**Figure 1 nutrients-14-04809-f001:**
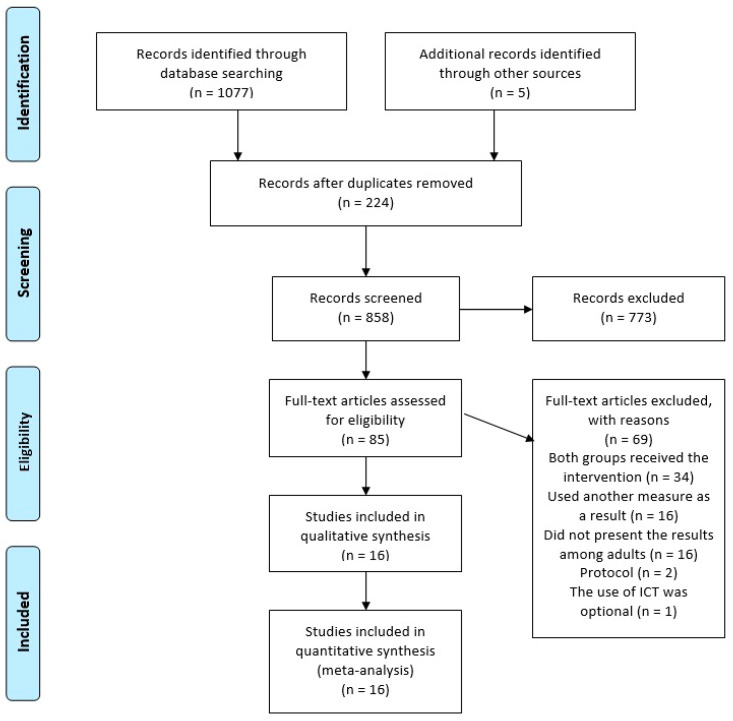
Flowchart of the eligibility of articles and final inclusion in the present study.

**Figure 2 nutrients-14-04809-f002:**
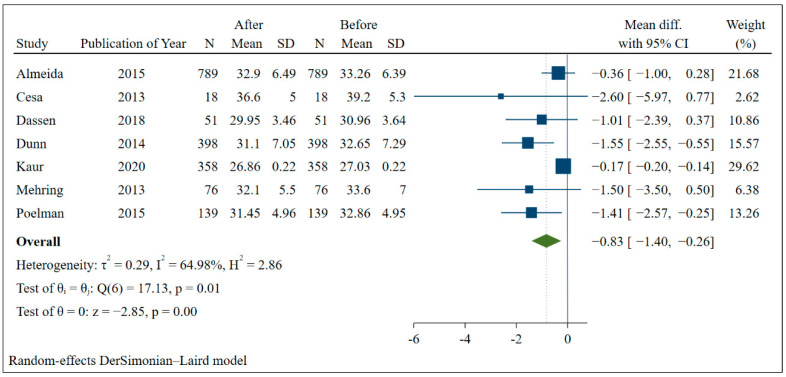
Meta-analysis of the mean difference in BMI of the intervention group and the control group.

**Figure 3 nutrients-14-04809-f003:**
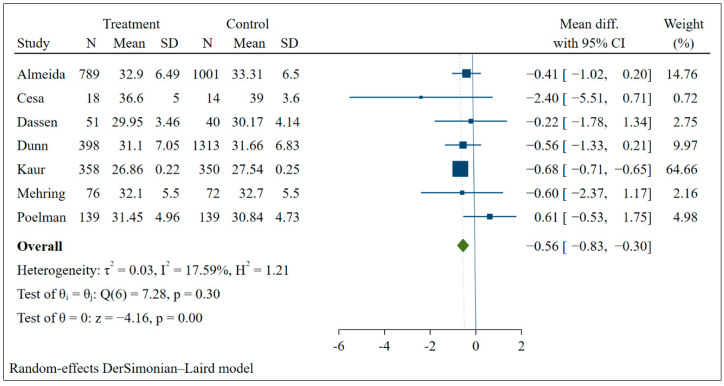
Meta-analysis of the mean difference in BMI before and after intervention.

**Figure 4 nutrients-14-04809-f004:**
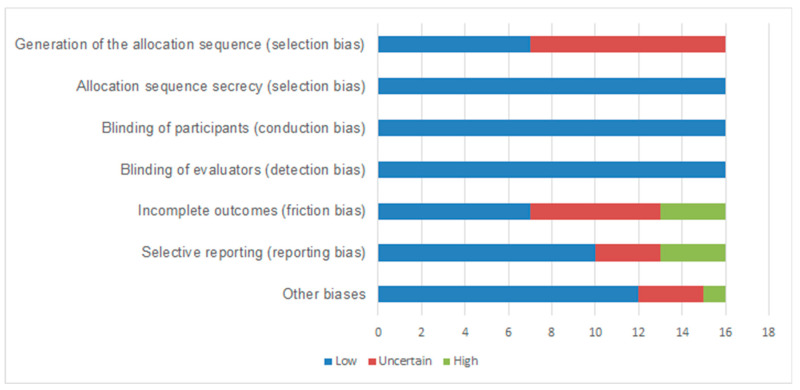
Risk of bias of the evaluated studies.

**Table 1 nutrients-14-04809-t001:** Characteristics from studies and change of BMI before and after intervention using Communication and Information Technologies.

First Author, Year	Country	n	Profile of Participants	Time in Months	Intervention *	Control	BMI Group in Intervention		BMI Group of Control	
							Before	After	Before	After
Lisón, 2020 [25]	Spain	105	Average age: control: 51.4; Intervention: 54.9 years old	3	In addition to the usual medical care, they had access to a webpage with documents and videos for download, interactive and self-instructive, with nine modules focusing on obesity and hypertension—eating habits and physical activity through self-monitoring, behavioral recording, stimulus control and techniques of problem solving.	Personal attendance	29.9 (SD: 2.6)	–0.4 (95% CI: –0.7; –0.2)	30.1 (SD: 2.7)	0.3 (95% CI: −0.5; 0.1)
Kaur, 2020 [16]	India	708	Average age: 52.7 years old. Gender: 76.1% women. Marital status: 89.3% married. Education: 51.2% with teaching higher.	6	Weekly sending of texts, images and video by SMS, e- mail, social networking application (WhatsApp). Website access ‘SMART Eating’ (recommendations, tips, BMI calculator answers the main doubts, games and among others), in addition to IT component, culinary calendar templates, plate composition education and measuring spoons.	They received printed material equal to the experimental group	27.03 (SD: 0.22)	26.86 (SD: 0.22)	27.45 (SD: 0.25)	27.54 (SD: 0.25)
Bennett, 2018 [23]	USA	351	average of age: 50.7. Ex: 32% men	12	Goal self-monitoring program of four behavioral goal changes. Use of voice response and messages, training calls with 10–15 min. The goals changed periodically.	Traditional treatment with self- monitoring materials	35.9 (SD: 4.1)	−1.4 (95% CI: −1.7; −1.1)	35.9 (3.7)	0.2 (95% CI: −0.07; 0.5)
Dassen, 2018 [24]	Netherlands	91	average age 47.9 years old. Gender: 74.7% women. Education: 52.7% had education higher	3	Memory game to improve capacity knowledge about self-control, eating style, eating disorders and healthy eating.	Training equal to online, only presential	30.96 (SD: 3.64)	29.95 (SD: 3.46)	30.49 (SD: 3.97)	30.17 (SD: 4.14)
Velázquez-Lopez, 2017 [18]	Mexico	351	average age control: 53.7 and intervention: 55.4 years old. sex: control 66.3% women and intervention 70.2% women	21	Multimedia program in Diabetes and Nutrition with modules of educational programs on diet, physical activity and various aspects of diabetes (causes, treatment and complications). The program has Written material, videos and exercises for content fixation.	Nutritional therapy, customized by a nutritionist	30.8 (SD: 5.9)	−0.42 (95% CI: −0.86; 0.01)	30.4 (SD: 5.0)	−0.07 (95% CI: −0.39; 0.25)
Tarraga Marcos, 2017 [28]	Spain	116	Gender: 58.6% women.	3	Multimedia program—access to adherence to Mediterranean Diet Screener and control of its adhesion level. Sending of the test every day (auto- monitoring and food diary) and the platform displayed the participant’s adherence weekly. In cases of low compliance, the platform offered personalized advice and suggestions (such as recipes etc.).	Oral and written information about health, food choices based on the Mediterranean diet and Exercises	30.8	29.3	30.7	30.1
Poelman, 2015 [26]	Netherlands	278	Average age: 45.7 years. Gender: 84.5% women. Education: 36.4% middle level	12	Multimedia program for portion control strategies in different environments and throughout the day.	traditional treatment and presential	32.86 (SD: 4.95)	31.45 (SD: 4.96)	32.00 (SD: 4.57)	30.84 (SD: 4.73)
Almeida, 2015 [21]	USA	1790	Average age: 47 years old	12	program daily delivery of tailored emails to each participant based on gender, fitness program, location, and barriers to promoting healthy eating.	educational sessions on job site	33.26 (SD: 6.39)	32.90 (SD: 6.49)	33.51 (SD: 6.44)	33.31 (SD: 6.50)
Watson, 2015 [27]	England	65	Average age: intervention 51.4 years old and control: 52.9 years. Gender: I50% women and control 61%.	3	Multimedia diabetes management program—planning, self-monitoring, goal setting and supportive feedback (physiologists) via email and phone with focus on good nutrition, physical activity and risk for cardiovascular disease.	Doctor care and guidelines about healthy life habits, model traditional and presential	32.9 (SD: 3.07)	−1.16 (95% CI: –1.60; –0.73)	32.4 (SD: 2.74)	–0.14 (95% CI: –0.47; 0.19)
Block, 2015 [22]	USA	341	Average age: 55 years. Sex: 68.7% men. Education: 82.9% teaching higher	6	Regular contact program and weekly definition of goals. Sending e-mails and reminders by cell phone. Weekly personalized program with goal setting to change health behavior—diet and physical activity.	conventional care through presential care.	31.1 (SD: 4.5)	−1.05 (95% CI: −1.06; 1.05)	31.2 (SD: 4.3)	−0.39 (95% CI: −0.39; −0.38)
Naimark, 2015 [17]	Israel	85	Average age: 47.9 years old Sex: 64% women	3	Application based on the web. The application allows users to monitor their food intake and physical activity, receiving real-time feedback. With base in systems theory control (CST) of regulation.	A talk about healthy habits	26.2 (SD: 3.9)	−0.48 (SD: 0.13)	25.0 (SD: 4.4)	−0.03 (SD: 0.12)
Dunn, 2014 [19]	USA	1711	Average in age: control 48.8 and intervention 49.1 years	19	Web conference: 15 weekly classes of one hour. 1 h sessions Elluminate Live teaching software	Same as Intervention, only presential.	32.65	31.31	32.64	31.66
Cesa, 2013 [31]	Italy	32	Average age: 31.8. Sex: just with women. Marital status: control 29.9% and intervention 32.9% married	12	Multimedia program with 14 virtual environments used by the therapist during a 60-min session with the patient. Environments where people eat and two body image comparison areas where patients practiced both food/emotional/relational management and general decision-making and problem-solving skills	Standard cognitive behavior therapy and face-to-face hospital treatment	39.2 (SD: 5.3)	36.9 (SD: 5)	41.1 (SD: 3.3)	38.3 (SD: 3.0)
Mehring, 2013 [30]	Germany The	148	Average age: 47.8 years old. Gender: 68.8% women.	3	Multimedia program—based on the principles of cognitive behavioral therapy. After a pre-assessment, the program generated an individual training based on the physicians’ recommendations, the physical characteristics and everyday behavior of the participants.	Individual counseling on usual care to reduce weight in the same way as the intervention group	33.6	32.1 (SD: −1.5)	33.3	32.7 (SD: −0.6)
Lachause, 2012 [20]	USA	312	Average age: 24.8 years. 75.64% women	3	An interactive web-based program provided nutrition and physical education information to college students. It had assessments to provide individual user feedback, information links (Ask the Expert, Student Voices, News) and four core learning modules.	Face-to-face course once a week for approximately 2 h for 12 weeks with the same topic as the online one.	29.5	28.75	29.19	29.0
Morgan, 2009 [29]	England	65	Average age: 35.9 years old. study only with men	6	A face-to-face information session (75 min) with instructions regarding diet modification, physical activity habits and behavior change strategies. Then they spent 3 months receiving online support through a website	Only participated in the face-to-face information session.	30.6	29 (SD −1.6; 95% CI: −2.2; −1.0)	30.5	29.4 (SD: −1.1; 95% CI: −0.5; 1.7)

* All included studies are randomized clinical trials; n—number of participants; 95% CI: 95% Confidence Interval; SD: Standard Deviation.

## Data Availability

The data of the present study can be obtained through correspondence with the indicated author.

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
