# Peer review of "Use of Information and Communication Technologies among Adults in Weight Control: Systematic Review and Meta-Analysis"

_nutrients, 2022, doi:10.3390/nu14224809_

Round 1

Reviewer 1 Report

This review conducted a review of the effect of 16 intervention studies that employed information or communication technologies on eating behaviors and obesity. The authors conducted a meta-analysis on the 8 studies with BMI. There was no mention of the effects on eating behaviors.

The interventions in the meta-analysis included diverse forms of technology, intervention techniques/procedures (some including human-delivered intervention), and durations. While a meta-analysis could be done on the 8 studies with BMI, what does it tell us? The results certainly don't mean that any study incorporating technology will have a desired effect. The authors should rethink and rewrite this manuscript as a scoping and/or narrative review of the 16 studies, perhaps within sub-groups with methods similar in strategically important ways.

Author Response

We welcome feedback from reviewers and suggestions.We will answer your questions here (the answers are highlighted). The suggestions of the third reviewer were incorporated into the manuscript and are highlighted in red  

Reviewer 1

The interventions in the meta-analysis included diverse forms of technology, intervention techniques/procedures (some including human-delivered intervention), and durations. While a meta-analysis could be done on the 8 studies with BMI, what does it tell us? The results certainly don't mean that any study incorporating technology will have a desired effect. The authors should rethink and rewrite this manuscript as a scoping and/or narrative review of the 16 studies, perhaps within sub-groups with methods similar in strategically important ways. We appreciate the comments. We believe that the meta-analysis should be maintained, as the use of information and communication technologies was used in all intervention groups and the evaluated outcome (body mass index) was present in all studies, thus enabling the verification of a summarized measure. We are grateful for the reviewer's comment, which made it possible for us to review our results and we are certain that it is possible to affirm that, considering the studies included, information and communication technologies (regardless of the intervention) contributed to the decrease in the body mass index.

Reviewer 2 Report

In this paper, the authors present a systematic review that evaluated the impact of information and communication technologies on changing eating habits, with the outcome being the change in body weight among adults.

This is a well-written interesting paper that contributes to the understanding of the therapeutic effect of educational technologies and evaluation of their impact on adult patients with obesity.

Author Response

We welcome feedback from reviewers and suggestions.We will answer your questions here (the answers are highlighted). The suggestions of the third reviewer were incorporated into the manuscript and are highlighted in red.

Reviewer 2

In this paper, the authors present a systematic review that evaluated the impact of information and communication technologies on changing eating habits, with the outcome being the change in body weight among adults. This is a well-written interesting paper that contributes to the understanding of the therapeutic effect of educational technologies and evaluation of their impact on adult patients with obesity.

We appreciate the evaluation and we also believe that our results can contribute to validate the use of information and communication technologies in the care of individuals who need to lose or control weight.

Reviewer 3 Report

Thank you. A useful meta-analysis. Overall it is well written. The research methodology is appropriate for the research questions but some minor changes is recommended to improve reporting.

Please consider the queries below:

Table 1 – should include study design column (to fulfil PICOS requirement)

Please provide justification for assuming a random-effects model to analyse the data

Was there publication bias which might be a threat against the results?

Figures – please include include a vertical line to cross unity so that the figures can be read more easily

Author Response

We welcome feedback from reviewers and suggestions.We will answer your questions here (the answers are highlighted). The suggestions of the third reviewer were incorporated into the manuscript and are highlighted in red.

Reviewer 3

Thank you. A useful meta-analysis. Overall it is well written. The research methodology is appropriate for the research questions but some minor changes is recommended to improve reporting.

Table 1 - must include the study design column (to meet the PICOS requirement) - We appreciate the suggestion. All included studies are randomized controlled trials as stated in the method section. We added a footnote to Table 1 indicating the study design.  

Provide justification for assuming a random effects model to analyze the data - The random effects model was chosen to adopt a more conservative approach, since statistical heterogeneity was considered high for one of the meta-analyses. Furthermore, high methodological heterogeneity was perceived between the studies. If another type of effect (fixed effect) was adopted, a distorted result could be produced, since a single study would have a weight of 99% in the final result of the meta-analyses. Additionally, we forward the results of the fixed effect for the evaluation of the evaluators (End of manuscript - afer references). We have included in the method section a description of the choice of model used.   

Was there publication bias that could be a threat to the results? There is a possibility of publication bias, but it was not possible to measure it due to the fact that the meta-analysis had fewer than 10 studies. We included in the method section that the measurement of publication bias was not performed due to the number of studies less than 10. And, in the discussion section we acknowledge that information bias may be present in this review.   

Figures – include a vertical line to cross the unit so figures can be read more easily. We appreciate the suggestions - we take the suggestion.

Round 2

Reviewer 1 Report

The authors were responsive to this reviewer's comments.